# Development and Validation of a Canine Health-Related Quality of Life Questionnaire and a Human–Canine Bond Questionnaire for Use in Veterinary Practice

**DOI:** 10.3390/ani13203255

**Published:** 2023-10-18

**Authors:** Robert P. Lavan, Muna Tahir, Christina O’Donnell, Alex Bellenger, Elodie de Bock, Patricia Koochaki

**Affiliations:** 1Merck Animal Health, Outcomes Research (CORE), Rahway, NJ 07065, USA; 2ICON plc, South County Business Park, Leopardstown, D18 FK72 Dublin, Ireland; munajtahir@gmail.com (M.T.); christina.odonnell@iconplc.com (C.O.); alex.bellenger@iconplc.com (A.B.); elodie.debock@iconplc.com (E.d.B.); patricia.koochaki@iconplc.com (P.K.)

**Keywords:** canine, health-related quality of life, human–canine bond, validity, reliability, outcomes

## Abstract

**Simple Summary:**

Two new tools designed to improve veterinarian-canine caretaker communications and lead to better health outcomes for the dog have been developed for canine caretakers to complete during visits to the veterinarian’s office. Qualitative and quantitative testing methods were used to develop valid and reliable new tools for dogs that measure a canine’s health-related quality of life (HRQoL-Q) and the character of the Human–Canine Animal Bond (HCBQ). Being reliable means that the veterinarian can count on the measures to provide consistent results (the results can be reproduced under the same conditions). Being valid means the tools have been shown to accurately measure what they are intended to measure. The Canine-HRQol and HCBQ can be used by veterinarians during office wellness visits to enhance communication between the pet owner and the veterinarian, help improve diagnosis of new problems that are of concern to the pet owner, monitor ongoing problems, monitor the relationship between the dog and caretaker that can impact the dog’s HRQoL, help the veterinarian and caretaker make shared decisions on treatment options, and document the health status of the dog.

**Abstract:**

The use of valid questionnaires to assess dogs’ health-related quality of life (HRQoL) in veterinary practice can improve canine health outcomes and communications between veterinarians and caretakers of dogs. The Canine HRQoL Questionnaire (Canine HRQoL-Q) and the Human–Canine Bond Questionnaire (HCBQ) were developed and validated to fulfill this need. A literature review, interviews with veterinarians, and focus groups with caretakers were conducted to generate questionnaire items and develop draft questionnaires, which were piloted with caretakers to establish their content validity. Measurement properties were evaluated using data from a prospective survey study (*N* = 327). Draft Canine HRQoL-Q and HCBQ measures were developed, including a domain structure, items, recall period, and scale/response options. Refinements were made via iterative cognitive interviews with caretakers. When no additional revisions were indicated and content validity was established, the questionnaires were psychometrically tested. Ceiling effects were observed for all items, and factor analyses indicated that the pre-specified domains are appropriate. Internal consistency was demonstrated for the HCBQ (α = 0.79–0.86) and all but the social functioning domain of the Canine HRQoL-Q (α = 0.60). Test–retest reliability for the Canine HRQoL-Q was generally moderate-to-good (with intraclass correlation coefficients (ICCs) > 0.79). Test–retest reliability for the HCBQ was moderate (ICCs: 0.70–0.79) except for the trust domain (ICC: 0.58). Known-groups validity was demonstrated via significant differences (*p* < 0.05) in scores for health/bonding groups. Convergent validity was supported (r > 0.40) between all domains and the total scores for both questionnaires. The Canine HRQoL-Q and the HCBQ are valid, reliable measures of canine HRQoL for use in veterinary clinics and appear to measure related but distinct concepts that contribute to canine health and wellness.

## 1. Introduction

Health-related quality of life (HRQOL) is multi-dimensional and includes concept related to physical, mental/cognitive, emotional, and social functioning and focuses on the impact health status has on quality of life. HRQoL measures can be either generic or disease-specific. A generic HRQoL measure consists of health-related items to assess the general health status and well-being. Disease-specific HRQoL measures include items specifically related to a disease state and are designed to be more sensitive to change when assessing the disease state. HRQoL measures are increasingly used in human medicine in clinical trials to assess the efficacy and safety of new medicines and in clinical care to track patients’ health status over time, encourage discussions between health care providers and patients, and support better health outcomes.

The development and use of HRQoL measures in assessing human health and in drug approval and clinical care has gained the support of regulatory agencies. The FDA guidance for the pharmaceutical industry on the development of valid and reliable outcome measures for use in human studies (2009) emphasizes the importance of qualitative research to the development of questionnaires that measure what is most important to patients in a population of interest and the use of quantitative research to evaluate the measurement properties of the questionnaires [1]. At present, the development of four Patient-Focused Drug Development guidance documents by the FDA is underway to guide stakeholders with respect to the collection and submission of patient experience data via HRQoL measures [2].

While the overall general health status of companion animals is the central focus of routine visits to the veterinary clinic, the availability of generic, valid, and reliable HRQoL measures developed to the standards required in human medicine to determine health status and track progression over time is limited. At the same time, the recognition of a key unmet need for HRQoL measures that foster communication between pet caretakers and veterinarians to guide decisions on routine care in clinical practice and monitor pet HRQoL over time is growing [3,4,5,6]. In human medicine, patient reported outcome (PRO) measures, traditionally used in clinical research, are increasingly being incorporated into clinical practice to capture patient experience and guide clinical decision making [7]. Further, a study aimed at determining if the use of a PRO measure as part of routine orthopedic clinical care is associated with improved patient experience showed that, when a PRO measure was included, patients were significantly more likely to feel that the provider had spent enough time with them, to recommend this provider office to another patient and to rate the provider significantly higher on a scale from 0 to 10 [8]. Other studies have also shown that systematically monitoring patients’ symptoms using HRQoL measures improves patient–clinician communication, clinician awareness of symptoms, symptom management, patient satisfaction, quality of life, and overall survival [9].

To date, most measures of canine HRQoL have been developed and utilized primarily to assess the impacts of specific conditions such as cancer and dermatitis or related concepts such as pain and pruritus [10,11]. While some generic measures of canine HRQoL do exist [12,13,14], they do not appear to follow all of the principles or best research practices in the regulatory guidance documents for the development of valid and reliable HRQoL measures. In particular, the guidelines stress the need to undertake qualitative research with the target population (caretakers of dogs) who will be completing the measure to understand and identify the concepts related to dogs’ HRQoL that they perceive as important to measure. The items for the measurements should be generated based on this understanding, using the language of the target population. Once the items are generated and a draft questionnaire is developed, the guidelines support piloting the measures via cognitive interviews with additional caretakers to revise the questionnaires and qualitatively establish content validity prior to quantitative testing [1,2].

According to the American Veterinary Medicine Association, “the human-animal bond is a mutually beneficial and dynamic relationship between people and animals that is influenced by behaviors essential to the health and wellbeing of both” [15]. There is recognition that a dog’s HRQoL is impacted by the bond between the caretaker and canine that is dependent on their relationship to each other and contributes to the HRQoL of both the canine and caretaker [16]. However, none of the existing canine HRQoL measures include an assessment of this aspect of canine HRQoL. Thus, development of a valid and reliable measure to assess the relationship between a caretaker and dog or a human–canine bond (HCB) could provide a more comprehensive, holistic understanding of canine HRQoL.

The current project was undertaken to fill the need for comprehensive, holistic measures based on concepts important to veterinarians and caretakers of dogs. The development and validation of two new, related outcome measures, the Canine HRQoL Questionnaire (Canine HRQoL-Q) and the Human–Canine Bond Questionnaire (HCBQ), developed in accordance with the principles outlined in the 2009 FDA guidance, are reported here.

## 2. Materials and Methods

The Canine HRQoL-Q and HCBQ were developed in two parts, using a mixed-methods approach. The first part of development included a targeted literature review and qualitative concept elicitation interviews with veterinarians and caretakers of dogs.

The literature review was conducted to identify the key concepts of canine HRQoL and the human–animal bond, as well as available questionnaires measuring canine HRQoL and the human–animal bond. Ninety-six articles were identified via OVID, thirteen of which underwent full-text review. An additional nine background articles were provided by Merck Animal Health and included in the review. The full-text review revealed HRQoL and human–animal bond concepts and informed questionnaire items for debriefing in the focus groups and interviews with caretakers of dogs.

Concept elicitation interviews were conducted with three practicing veterinarians. The objectives of these interviews were to identify and understand the concepts that veterinarians consider important to measure to determine the strength of the bond between a dog owner and their dog and the concepts that veterinarians consider important to measure to determine the HRQoL of a dog.

A series of four focus groups was conducted with a convenience sample of caretakers of dogs (*N* = 20 caretakers of dogs) residing in the US who were recruited from ICON plc, a global clinical research organization. The objectives of these interviews were similar to those for the interviews with veterinarians. Results from the literature review and interviews were used for the development of items that were reflective of the descriptions and language used by caretakers and to define a domain structure for the questionnaires. In addition to the items, a recall period (the time that should be considered when responding to items) and a scale were chosen that were appropriate for the items, leading to the development of the two draft questionnaires, the Canine HRQoL-Q and the HCBQ.

Once the draft questionnaires were developed, cognitive interviews were conducted with a separate group of caretakers of dogs to establish the content validity of the questionnaires and to refine the questionnaires for quantitative psychometric testing. For content validity to be established, (1) each item in the questionnaire should have a single concept or express a single idea to caretakers and be unambiguous and easily comprehended by the target audience; (2) there should be no important missing concepts; (3) the instructions for completion and the recall period should be understood and appropriate; and (4) it should be easy for respondents to choose a response to each item (the scale, number, and type of response options are appropriate for the items) [17]. The interviews were conducted iteratively to allow the research team to evaluate the results and make indicated revisions as the interviews progressed. The interviews were conducted until no additional revisions were indicated.

The second part of the study focused on the quantitative assessment of the psychometric properties of the Canine HRQoL-Q and the HCBQ (i.e., the validation of the measures). The measurement properties assessed included item-level analyses, tests of reliability (internal consistency and test–retest), and an assessment of validity (factor structure, known-groups validity, and convergent validity).

### 2.1. Part 1: Questionnaire Development and Content Validation

#### 2.1.1. Concept Elicitation, Item Generation, and Draft Questionnaire Development

Concept elicitation to identify concepts important to canine HRQoL and the HCB was undertaken with 20 English-speaking caretakers of dogs in the US who cared for a diverse population of dogs with respect to age, size, breed, and varying health status. The participants were included if they were the caretaker of a dog for at least six months. Caretakers of dogs were excluded if the dog was a foster animal, as they may not have known as much about the dog’s HRQoL or may not have bonded with a dog similarly to an owner. The participants were sampled from ICON plc employees located at various locations across the US between February and March of 2021. Four virtual focus groups with approximately five participants each (twenty in total) lasting up to 120 min were conducted in English via a WebEx video-enabled platform. The ICON employees were purposively selected to obtain as diverse a population of participants as possible with respect to gender, age, ethnicity, geographic location, and the breed/health status of dogs.

The participants were required to create photographic collages prior to joining the focus groups. Photographic collage is a projective technique in which participants project their opinions and beliefs onto images that they select with respect to a particular topic. This allows participants to articulate their thoughts and experiences at a subconscious level that is difficult to reach with the direct questioning techniques frequently used in qualitative interviews [18,19,20]. For the focus groups, the participants were asked to choose several photographs and write a brief description explaining why they chose each photograph to create two photographic collages, one collage depicting the concepts they perceived as important to a dog’s HRQoL and a second collage for the HCB. During the focus groups, the participants elaborated on these concepts, explaining why they selected each image, which enabled them to articulate and expand upon how and why they perceived the concepts elicited via the photographs to be important to them. The order of the discussion of the HRQoL and HCB collages was counterbalanced from group to group.

The participants were asked to complete the CHQLS-15, a 15-item canine HRQoL survey [12], and to provide perspective on the relevance and importance of the concepts addressed in the questionnaire. In addition, the participants reviewed and discussed 10–15 HCB concepts that had been developed from the review of the literature and veterinarian interviews and emailed to them prior to the focus group. They were asked to rate the concepts by ranking them as “important,” “maybe/maybe not important,” and “not important” based on how relevant each item was to their relationship with their dog(s). The items chosen by the participants as the most relevant were reviewed and discussed further. 

The results of the concept elicitation focus groups, literature review, and interviews with veterinarians informed the generation of items for the draft questionnaires. General considerations for item generation included the following: each item should contain a single concept, should be unambiguous and clearly stated in language understood by the respondent, and should contain concepts important or relevant to the respondents. Other elements of the questionnaire that were carefully considered included the instructions for completing the questionnaires, the recall period, and the response scale (i.e., the number of response options and the type of scale, e.g., frequency, agreement, and severity).

#### 2.1.2. Pilot Testing/Cognitive Interviews

Following the development of the first drafts of the questionnaires, cognitive interviews were conducted with a new group of caretakers of dogs to assess the content validity of the questionnaires, including the comprehension, interpretation, and relevance of each item, the ease of completion, the understanding of the instructions for both questionnaires, the utility of the scale, the appropriateness of the type and number of response options in the scale, and missing concepts.

Individual cognitive interviews were conducted with 16 English-speaking caretakers of dogs in the US. Similar to the concept elicitation focus groups, caretakers were included if they cared for a dog for at least six months, and caretakers were excluded if the dog was a foster animal. The interviews were conducted virtually using a video-enabled WebEx platform. Participants were recruited by the Schlesinger Group, a third-party recruitment agency, for interviews that were conducted in July 2021.

During the cognitive interviews, the participants were asked to complete the questionnaires and were then asked semi-structured questions regarding the ease of completing the questionnaires and their understanding of the instructions for both questionnaires, the relevancy and appropriateness of the recall period, their interpretation of each item and their perception of its relevance, the interpretability and suitability of the response options or scale, the ease of responding or choosing between response options, the number and type of response options, and any missing items or concepts that should be included. The order of the completion and discussion of the HRQoL and HCB questionnaires was counterbalanced. The questionnaires were refined based on responses from the participants, and the questionnaires were finalized for psychometric testing.

### 2.2. Part 2: Psychometric Validation

#### Study Population

The psychometric validation of the Canine HRQoL-Q and the HCBQ developed in the first part of the study was undertaken using data from a non-interventional, prospective survey with a 2-week follow-up period among caretakers of canines who provide at least 50% of a dog’s care (*N* = 327). The participants included caretakers of dogs for at least six months who were at least 18 years of age, spoke English, lived in the US, and were able to provide consent to participate in the study. Caretakers of foster dogs were excluded from the study. Data were collected using online surveys. Interested participants received an email link to determine their eligibility. If participants met the eligibility criteria, they were able to review the study information sheet and consent to participating in the study. Following screening and the collection of informed consent, the participants were asked to complete a sociodemographic form and were redirected to the survey (which included the Canine HRQoL-Q and the HCBQ). An Owner’s Global Impression of Health (OGIH, a single-item measure that assesses a caretaker’s impression of their dog’s general health as “excellent,” “very good,” “good,” “fair,” or “poor”) was also assessed. The participants who were followed- up with were only required to complete the Canine-HRQol and HCBQ questionnaires at the second time point/visit.

### 2.3. Statistical Analysis

SAS version 9.4 (SAS Institute, Cary, NC, USA) was used for all statistical analyses. The analyses of the psychometric properties of the Canine HRQoL-Q and the HCBQ were conducted using data from all caretakers included in the psychometric study dataset, as described in Section 2.3.1, Section 2.3.2, Section 2.3.3 and Section 2.3.4. The electronic collection system required the respondent to complete each item to proceed through the questionnaire; skipping items was not permitted, and no missing values were expected. Responses from incomplete questionnaires were not used. Missing scores on the OGIH were not imputed for the psychometric validation as this is a single-item measure.

#### 2.3.1. Item-Level Analyses

Item-level analyses were evaluated using visit 1 data and included the use of response categories for each item (i.e., the frequency and percentage of participants in each response category), measures of central tendency (to assess the distribution of total and domain scores), and an assessment of floor and ceiling effects [21].

#### 2.3.2. Factor Structure

A confirmatory factor analysis (CFA) was performed at visit 1 to evaluate the factor structure of the Canine HRQoL-Q and the HCBQ as the measures were developed with pre-specified domains. Maximum likelihood estimators with robust standard errors were used to identify the factor solution, with the following fit statistics computed: the Comparative Fit Index (CFI), the Root-Mean-Square Error of Approximation (RMSEA) with a 95% confidence interval (CI), and the Standardized Root Mean Squared Residual (SRMR).

#### 2.3.3. Reliability

Internal consistency reliability was evaluated at visit 1 using Cronbach’s alpha coefficient, with a target value of 0.7 indicating good internal consistency, and using item–total correlations, with a significant correlation >0.30 showing good homogeneity [22,23,24].

Test–retest reliability was assessed between visits 1 and 2 among dogs whose owners reported no change in the OGIH (i.e., stable dogs). Mean differences were calculated to compare Canine HRQoL-Q and HCBQ total and domain scores between the two assessment visits. The intraclass correlation coefficients (ICCs) with 95% confidence intervals were computed where ≥0.7 (absolute ICC and lower 95% confidence interval limit) indicated good reproducibility [25,26,27].

#### 2.3.4. Validity

Known-groups validity was examined at visit 1 to determine whether the Canine HRQoL-Q and HCBQ can distinguish between groups with expected differences in scores. For the Canine HRQoL-Q, the groups were defined using (1) canine health/disease state (healthy, non-food allergy/skin problem, and ear infection); (2) OGIH groups (excellent, very good, good, and fair/poor); and (3) general health/HRQoL item tertiles (items 8a and 8b on the Canine HRQoL-Q). For the HCBQ, the groups were defined using (1) canine health/disease state (healthy, non-food allergy/skin problem, and ear infection); (2) OGIH groups (excellent, very good, good, and fair/poor); and (3) general bonding item tertiles (items 4a and 4b on the HCBQ), although the former categories may not work as well for bonding. Using analysis of variance (ANOVA) models, differences in the total and domain scores for both questionnaires were assessed by groups of severity. Although the questionnaires are intended to be used with healthy dogs, the recruitment for this study included dogs with variable health statuses as reported by their caretakers (i.e., those that were considered healthy by their caretakers as well as dogs that were reported to be unwell, specifically those with atopy and ear infections) to assess this psychometric property.

Given that no other measures were included in the study, convergent validity was assessed by examining correlations between all total and domain scores for the Canine HRQoL Questionnaire and HCBQ separately (i.e., using a domain–domain type matrix) at visit 1. Stronger correlations were expected where larger overlaps of concepts were evident, thus suggesting that the tools measure as intended. Further, the total scores of the Canine HRQoL-Q and HCBQ scores were correlated with each other (using Pearson correlations) to test whether the HCB is associated with canine HRQoL.

## 3. Ethical Considerations

Ethical approval was obtained from Salus IRB, a central Ethics Committee, for both Parts 1 and 2 of the study.

## 4. Results

### 4.1. Part 1: Qualitative Development of Draft Versions of the Canine HRQoL-Q and HCBQ

#### 4.1.1. Development of Initial Drafts of the Canine HRQoL and HCB Questionnaires

The baseline characteristics of the participants included in the focus groups are presented in Appendix A. The participants were predominantly White (80%) and female (80%), with a mean age of 45 years. Their dogs were diverse with respect to age, breed, size, and health status. The mean age of the dogs was 7.3 years (see Appendix A). The highest proportions of participants reported owning their dog for between 1 and 5 years (44%) and between 5 and 10 years (32%). Most caretakers reported visiting the vet for their dogs every 3–6 months (24%), twice per year (20%), and once per year (36%). Dog breeds varied and included a number of mixed breeds (e.g., Schnoodle, Goldendoodle, Pomeranian–Chihuahua, Yorkshire Terrier mix, and Poodle mix). No participants reported that their dog was a service dog; however, some dogs received therapy dog training (4%), citizen training (4%) (i.e., training involving the teaching of good manners and responsible ownership of dogs), or training as an emotional support animal (4%). Finally, most dogs were reported to be generally healthy (68%).

The HRQoL concepts identified for inclusion in the draft Canine HRQoL-Q included a healthy weight, mobility, exercise, spending time with the family, everyday/frequent stimulation, mood, health indicators, food and water, poor appetite, age-related issues, training, mental status, and separation. From the collages, HRQoL discussion, and the completion of the CHQLS-15, the concepts important to caretakers were identified and grouped into domains or themes and items reflective of the concepts and language used by the caretakers were generated for the development of the draft Canine HRQoL-Q.

The draft Canine HRQoL-Q consisted of thirty-six items organized into eight domains or themes: mobility, energy and vitality, physical health, appetite and hydration, emotional functioning, cognitive functioning, social functioning, and general health. Each of the items was rated on a five-point Likert Scale as follows: 1 = strongly disagree; 2 = disagree 3 = neither agree nor disagree; 4 = agree; and 5 = strongly agree. The general health domain included two global concept items which assessed canine overall health and HRQoL, respectively, on a scale ranging from 0 (very poor) to 10 (excellent). Higher scores denoted a better canine HRQoL. The caretakers were asked to consider the items at the present time, in their current state.

During the HCB rating exercise, the participants rated several concepts as the most important, including meeting the dog’s needs (*n* = 15) (e.g., basic needs, enrichment needs, and social needs), caring for and protecting the dog (*n* = 14), the dog is a friend or companion (*n* = 14), the dog is a family member (*n* = 12), the dog is a priority in the caretaker’s life (*n* = 12), including the dog in activities (*n* = 11) (e.g., hiking and walking), recognizing the dog’s emotions (*n* = 11), the emotional benefits of caring for a dog (*n* = 10) (e.g., increased self-esteem), the caretaker as the leader (*n* = 10), caring for the dog in their old age (*n* = 15), caring for an unhealthy dog (*n* = 14), and the cost of caring for the dog (*n* = 10). The latter three concepts were end-of-life considerations; thus, they were not included in the draft questionnaire. The concept of the dog caring for and protecting the caretaker was rated with medium importance (*n* = 12). Based on these findings, the following domains or themes were identified: trust, communication, spending quality time/companionship, and security/comfort.

Based on the results from the focus groups, a draft HCBQ was developed as a 21-item measure that evaluates the human–canine relationship from the caretaker’s and dog’s perspectives across four domains (trust and security, communication, spending quality time/companionship, and general bonding). Similar to the Canine HRQoL-Q, each item assesses the degree to which the caretaker agrees or disagrees with statements regarding their relationship with their canine on a five-point Likert scale (1–5, ranging from 1 = strongly disagree to 5 = strongly agree). The general bonding domain included two global concept items which assessed the strength of the caretaker’s bond to the dog and the attachment of the caretaker to the dog on a scale of 1 (not strong/attached at all) to 10 (as strong/attached as I could imagine). Higher scores indicate a higher (stronger) human–canine bond.

#### 4.1.2. Assessing Content Validity and Refining the Questionnaires via Cognitive Interviews

The baseline characteristics of the participants included in the cognitive interviews are presented in Appendix A. The participants were predominantly female (56%) with a mean age of 49 years. Half of the participants were White (50%), and approximately one-third were married (38%) and one-third single (31%). Most participants worked either full-time (56%) or part-time (17%). Regarding their education level, 25% had completed some college education without a degree, 13% had a two-year associate’s degree, 31% had a four-year bachelor’s degree, 24% had a master’s degree, and 6% had a doctoral or professional degree. Most participants earned incomes in the USD 30,000–USD 59,999 (31%) and USD 60,000–USD 89,999 (31%) ranges. Lastly, the majority of participants (69%) did not have children in their household.

After completing each of the questionnaires, the participants’ comments and thoughts about the questionnaires were carefully reviewed and used to guide revisions to the draft questionnaires. Specifically, the participants provided information on the wording, clarity, and relevancy of the items, the recall period, the response options and scale, and the instructions that was used to edit each element of the questionnaires.

Based on comments from the caretakers, a number of items were removed from the HRQoL-Q due to a lack of relevance (e.g., “My dog liked to meet other people/dogs he/she does not know” was irrelevant as the participants reported that their dogs did not like strange dogs or people); a lack of universality (e.g., “My dog had difficulty climbing stairs” was not applicable to all dogs as some caretakers did not have stairs); multiple interpretations or meanings of some concepts (e.g., “My dog preferred to be alone” could be due to other factors such as the dog’s mood or the weather); overlapping or redundant items (e.g., “My dog played” and “My dog was interested in play”); a lack of clarity or comprehension (e.g., “frisk”, “pep”, “boundless energy”, “patchy coat”, “clear eyes”, and “easily confused”); confusion between similar items (e.g., “hair loss” and “shedding”); and the inapplicability of concepts to dogs (e.g., it was unclear what “easy-going” means for a dog).

Items in the HRQoL-Q were reworded due to inconsistent tense and phrasing (e.g., “My dog has no difficulty walking” changed to “My dog has difficulty walking”); problematic item interpretation (e.g., the interpretation of “stiff” leading to a change from “My dog’s movements were stiff” to “My dog has been limping”); breed-dependent concepts (e.g., “My dog was energetic” changed to “My dog’s energy level was the same as usual” as some breeds are not energetic); an overlap of concepts (e.g., “remembered where things were,” “was forgetful,” and “got lost somewhere that should have been familiar”); and a lack of clarity (e.g., “My dog did not recognize people or situations that should have been familiar to him/her” changed to “My dog seems to be forgetting people or places”).

Regarding the HCBQ, items were removed due to a lack of relevance (e.g., “My dog’s daily behavior is predictable” in the communication domain was irrelevant as unpredictable behavior did not impact communication); social desirability bias (e.g., “Trust is important for a good relationship with my dog”); an inability to measure the concept (e.g., participant could not tell if their “dog knows [they] really care about him/her”); the reflection of the caretaker (e.g., “I think my dog knows I really care about him/her” could be a reflection of the caretaker); overlap with other concepts (e.g., “My dog is soothed by my presence” overlaps with “My dog is comforted by my presence”); problematic item interpretation (e.g., “mental stimulation”, following owner “guidance”); preference for one concept over another (e.g., “companion” versus “inseparable”); and the context of the future administration of the questionnaire (e.g., “My dog would rather be with me than anyone else” does not fit the context of future administration as the person bringing the dog to the vet/completing the questionnaire is not necessarily the dog’s favorite person). 

Based on comments from the caretakers, some items in the HCBQ were reworded due to the relevancy of a concept (e.g., “My dog checks in on me throughout the day” changed to “My dog checks in on me routinely when we are together” as some people are not home throughout the day); inappropriate or awkward terminology (e.g., “My dog wants to make me proud of him/her” changed to “My dog seeks my approval”); and suggestions for more generally relevant phrasing (e.g., “My dog and I enjoy exercising together” changed to “My dog and I enjoy physical activities together” since not all caretakers exercise).

After the completion of the cognitive interviews, the domain structure for the HRQoL questionnaire remained the same, whereas the domains for the HCB questionnaire changed somewhat. “Trust” and “Security/Comfort” were combined, resulting in a domain of “Trust & Security”.

Once the questionnaires were revised into the final draft versions, psychometric testing was undertaken.

### 4.2. Part 2: Psychometric Validation

#### 4.2.1. Sample Characteristics

The baseline characteristics of the participants included in the study are presented in Appendix A. The participants were predominantly White females (>60%) with a mean age of 48 years. The majority of participants (>60%) reported having no children in the household and owning only one dog. The highest proportion of participants were married (49.8%), working full- time (46.8%), and had a four-year bachelor’s degree or higher (~40%) with an income level of USD 30,000–89,999 (~51%).

The dogs on which the survey answers were based had a mean age of 7.1 years and were owned for an average of 6.2 years. The highest proportion of participants reported taking their dog to the vet once (41%) or twice (35%) a year. Few participants reported that their dog was a service dog (5.2%). The most commonly reported breed of dog was “mixed” (47%). Finally, most dogs were reported to be generally healthy (68%) (i.e., considered healthy by their owners), followed by those with skin problems (21%), non-food allergies (16%), and ear problems (16%).

#### 4.2.2. Item-Level Analyses

All items across both questionnaires exhibited ceiling effects such that ≥20% of the sample selected the best possible score. The Canine HRQoL-Q and HCBQ total scores had mean (SD) values of 8.00 (1.59) and 8.73 (1.15), respectively.

#### 4.2.3. Factor Structure

All items loaded onto their respective domains with factor loadings at or above 0.40. The fit statistics further supported these findings, with a CFI of 0.95, an RMSEA of 0.05, and an SRMR of 0.05.

#### 4.2.4. Reliability

Internal consistency reliability was demonstrated for the Canine HRQoL-Q, with domain-level Cronbach’s alpha values ranging between 0.81 and 0.86 for the energy and vitality, physical health, appetite and hydration, and emotional functioning domains, as shown in Table 1. The mobility and cognitive function domains had a Cronbach’s alpha > 0.90. Cronbach’s alpha for the social functioning domain was 0.60, with Cronbach’s alpha values for all items falling below <0.60. The domain-level Cronbach’s alpha values for the HCBQ ranged between 0.79 and 0.86, suggesting that the measure is internally consistent (Table 2). The item-level Cronbach’s alpha values were all >0.70, with the exception of item 2d (α = 0.69). Further, all Canine HRQoL-Q and HCBQ items had moderate to strong correlations (r = 0.42–0.84) with the remaining items’ total score (i.e., they met the pre-specified threshold of 0.30) (Table 3 and Table 4, respectively).

For the Canine HRQoL-Q, test–retest reliability was demonstrated among *n* = 54 dogs whose owners reported no change in the OGIH value for the total score and the mobility and physical health domains, with an ICC value of >0.80 (Table 5). Further, the ICC values for the energy and vitality, emotional functioning, and social functioning domains were acceptable-to-good (0.79–0.80), although the 95% confidence intervals were wider, with a 95% lower bound of <0.70. Somewhat lower ICCs were observed for the appetite and hydration and cognitive functioning domains (0.66 and 0.65, respectively), with 95% lower bounds of 0.47 and 0.46, respectively, suggesting poor reproducibility for these domains. There were minimal score changes between visits 1 and 2 for all scores. Test–retest reliability for the HCBQ was moderate overall, as demonstrated in Table 6. While only small decreases in the total and domain scores were observed between visits 1 and 2, ICCs between 0.70 and 0.77 (with 95% confidence interval lower bounds <0.70) were demonstrated for the total score and the communication and quality-time domains. The trust domain demonstrated poor reproducibility (ICC: 0.58; 95% confidence interval: 0.37–0.73).

#### 4.2.5. Validity

Regarding known-groups validity, the Canine HRQoL-Q was able to differentiate between dogs (*p* < 0.01) based on canine health/disease state, OGIH ratings, and canine general health and HRQoL (Table 7). Specifically, significant differences (*p* < 0.001) in the total score and the mobility, energy and vitality, physical health, and emotional functioning domains were observed at visit 1 between groups defined by canine health/disease states in the expected direction (i.e., healthy dogs had higher (better) scores compared to dogs with non-food allergies/skin problems and ear infections). No significant differences were observed for the appetite and hydration, social functioning, and cognitive functioning domains. When the Canine HRQoL Questionnaire scores were compared across the OGIH group categories, it was evident that the dogs whose owners reported them being in “excellent” health had significantly higher (better) scores compared to those in “poor/fair” health. Finally, the dogs with better health and higher HRQoL scores per the general health and quality of life items on the Canine HRQoL Questionnaire had significantly higher (better) scores compared to those in poor health/with poor quality of life. Strong effect sizes were shown for the OGIH and the Canine HRQoL Questionnaire’s general health and bonding items for all domain scores and for the total score. Similar findings were observed for the HCBQ (Table 8) such that the total and domain scores were significantly different between OGIH groups and by the general bonding items (with higher scores observed among dogs whose owners reported them being in excellent health and with a higher degree of bonding). However, there were no significant differences in scores between canine health/disease state groups.

The matrix of inter-correlations of the Canine HRQoL-Q total/domain scores and the HCBQ total/domain scores are shown in Table 9 and Table 10. Moderate-to-strong correlations were observed between all the Canine HRQoL Questionnaire domains, with especially strong correlations observed between the mobility and energy/vitality domains (r = 0.73) and between the physical health and both the appetite and hydration (r = 0.70) and cognitive functioning (r = 0.72) domains. Interestingly, the correlations between canine HRQoL domains associated with physical attributes (the mobility, energy/vitality, and physical domains) and the domains associated more closely with behavioral attributes (emotional, social, and cognitive functioning) were moderately strong, ranging from r = 0.52 to r = 0.72, underscoring the relationship between physical health and behavior. Strong correlations were also observed between the total score and all but the general health domain. For the HCBQ, a strong correlation was observed between the trust/security domain and both the communication (r = 0.74) and quality time (r = 0.75) domains. Further, a strong correlation of r = 0.75 was demonstrated between the communication domain and the quality-time domain. Similar to the Canine HRQoL-Q, very strong correlations (r > 0.87) were observed between each of the HCBQ domains and the total score, with the exception of the general bonding domain (r = 0.64). Overall, these findings provide strong support for the convergent validity of the measures. Finally, the Canine HRQoL-Q was moderately correlated with the HCBQ (r = 0.44).

## 5. Discussion

Published studies have shown that the health and well-being of companion animals can be better understood and health issues identified if there are assessment tools in the veterinary clinic that can be completed by caretakers of canines and shared with the veterinarian. For example, through an online questionnaire, Hale, et al. [28] found that almost all dog owners in the UK (95.8%) were comfortable discussing their dog’s quality of life with their vets, yet only a third of owners (32%) reported this topic was raised by their veterinarians. Furthermore, most owners (70.8%) were interested in accessing tools to assess their dog’s quality of life, but very few had experienced any form of formal health or well-being assessment tool (4.4%) with their veterinarians. 

Several disease-specific questionnaires, including questionnaires assessing pain, osteoarthritis, and cognitive dysfunction syndrome due to aging, and generic/general quality of life questionnaires have been developed for dogs. The DISHAA (Disorientation, Social Interactions, Sleep/wake Cycles, House-soiling, Learning, Memory, Activity, and Anxiety) questionnaire is available for the assessment of several domains associated with cognitive dysfunction syndrome in dogs [29]. Two of the most widely used tools to assess osteoarthritis in dogs are the Canine Brief Pain Inventory (CBPI) and the Liverpool Osteoarthritis in Dogs (LOAD) [30,31,32,33] assessment. The CBPI encompasses two sections: a pain severity score (PSS) and a pain interference score (PIS). The first assesses the magnitude of pain of an animal and the second assesses the degree to which the pain affects daily activities [34]. The Canine Orthopaedic Index (COI) was developed to assess four dimensions of OA in dogs: stiffness, gait, function, and quality of life [35]. The Hudson Visual Analogue Scale (HVAS) was compared with force plate analyses and was shown to be repeatable and valid for assessing the degree of mild-to-moderate lameness [36]. The Glasgow University health-related dog behavior questionnaire (GUVQuest), the first questionnaire developed for use in animals to measure chronic pain in dogs through its effect on HRQL [37], underwent validation in dogs with painful conditions, including degenerative joint disease. The GUVQuest consists of 109 items and four domains (mobility, physical appearance, and mood) and requires 30 min to complete. Subsequently, a shortened, generic version of the assessment, known as the Vetmetrica, was developed to measure HRQoL more generally in dogs via the selection of a core set of items from the GUVQuest. This new questionnaire consists of 46 items and the same four domains [38]. Subsequently, the 46-item Vetmetrica was further reduced to yield a new web-based, generic questionnaire comprising 22 items with four health-related quality of life domains (energetic/enthusiastic, happy/content, active/comfortable, and calm/relaxed) [39]. In 2022, the development and initial validation of a canine quality of life questionnaire, specifically designed to reliably quantify the full range of wellbeing in the general dog population, was reported. The questionnaire consists of 32 items, five daytime domains (energetic, mobile, relaxed, happy, and sociable), and three mealtime domains (relaxed, interested, and satisfied). The initial item set was developed by a five-person team consisting of pet owners, veterinarians, veterinary nutritionists, and veterinary behaviorists and a literature review. A survey methodology was used to select items for validity and reliability testing [14]. 

Comparing existing questionnaires with the Canine HRQoL-Q and HCBQ demonstrates that these questionnaires assess many of the same domains and concepts and are arguably more comprehensive than any of the existing questionnaires alone. The Canine HRQoL-Q and HCBQ measure concepts related to pain, mobility, energy/vitality, and emotional, social, and cognitive function, as well as assessing the relationship and bonding between dogs and their caretakers. This comprehensiveness, coupled with the use of the principals outlined in FDA guidelines for the development of valid, reliable observer-reported outcome measures and the experience and expertise of the developers in developing the Canine HRQoL-Q and HCBQ, provide an excellent option for use in veterinary practice. 

This study aimed to help fill this gap with the development and validation of two questionnaires, a Canine HRQoL Questionnaire and a Human–Canine Bond Questionnaire, to assess the overall health and well-being of dogs for use in veterinary clinics. These questionnaires aim to encourage discussions, alert the veterinarian and pet caretaker to potential health issues, and support decision making with respect to recommended actions, e.g., treatment, dietary changes, the need for further testing, end-of-life decisions, and improving veterinary care. The development of two questionnaires, one focused directly on health-related quality of life and a second focused on the bond between a caretaker and their dog, was undertaken based on the existing literature that supports the hypothesis that while the bond is a separate construct from HRQoL, the bond is related to and has an impact on the health-related quality of life of the canine.

By adopting established regulatory and industry guidelines to support the development of clinical outcome assessment measures for human drug approval, two valid and reliable measures were developed that can be used in a single veterinary clinic to improve health outcomes and monitor the health and well-being of its canine patients over time or across multiple veterinary clinics to compare canine health and well-being in a variety of settings or geographies. Data obtained from individual evaluations can be aggregated to establish normative population data to help veterinarians and caretakers understand how a dog’s scores compare with others in the database and identify deficits in the dog’s HRQoL that should be investigated further.

The development of valid and reliable questionnaires began with understanding what to measure from the perspectives of caretakers of dogs, veterinarians, and the existing literature. Interviews with veterinarians and focus groups comprising caretakers of dogs, combined with the use of multiple modes of qualitative data collection (e.g., photographic collages and a ranking task), provided an in-depth understanding of caretakers’ experiences with their dogs and helped identify the concepts important to caretakers for assessing a dog’s HRQoL, as well as the bond between the caretaker and dog. The results were used to develop a preliminary domain structure and generate items for the two initial draft questionnaires. Based on the content of the items and the caretakers’ descriptions of their experiences with their dogs, a preliminary recall period, scale, and response options were selected for the draft questionnaires.

Once the draft questionnaires were developed, individual cognitive interviews with additional caretakers of dogs were conducted to assess the questionnaires’ content validity and refine the questionnaires when issues were identified. The interviews proceeded until no additional revisions were indicated and content validity was established.

Based on the cognitive interviews/input from caretakers of dogs, a few items in the draft questionnaires were moved from one domain to another, removed, or reworded. Two domains in the draft HCBQ (i.e., “Trust” and “Security/Comfort”) were combined into a single “Trust and Security” domain for reasons of efficiency and the similarity of the concepts perceived to be relevant to these domains.

Following the demonstration of content validity, quantitative, psychometric testing of the performance of both the Canine HRQoL-Q and the HCBQ was undertaken in a non-interventional prospective survey study with participants recruited using similar criteria as in the qualitative research phase. The performance testing of both questionnaires indicated the presence of ceiling effects such that >20% of the caretakers of dogs selected the best possible score. These findings are expected given that the dogs on which the survey was based were predominantly healthy. Thus, a high percentage of respondents may have accurately reported that their dog did not experience any issues. The average scores for the Canine HRQoL-Q and the HCBQ (8.00 and 8.73, respectively) were close to the maximum possible score of 10, suggesting that the dogs in the study generally had a high quality of life and bonded well with their caretaker. The pre-specified factor structure of both the Canine HRQoL- Q and the HCBQ was also confirmed using a factor analysis, although some items exhibited lower factor loadings. Such items were considered for removal in unison with findings from other psychometric analyses.

Upon additional testing, the Canine HRQoL-Q demonstrated strong internal consistency on all but the social functioning domain. This could indicate that the social functioning items may not be contributing useful information to the Canine HRQoL-Q; thus, this domain was considered for removal from the measure. Further, the mobility and cognitive functioning domains each had a Cronbach’s alpha value > 0.90, indicating that some items within each of these domains may be redundant and can be eliminated from the measure. Item-total correlations were moderate-to-strong for all items in the Canine HRQoL-Q. The HCBQ was also shown to be internally consistent. Taken together, these findings showed that items within the measures generally reflect a single underlying construct and consistent item responses. 

The reproducibility of the Canine HRQoL-Q was strong, as evidenced by an ICC >0.79 and a minimal change in scores between visits 1 and 2. This is a strong result considering the relatively small subsample of dogs available for the analysis (*n* = 54) who did not exhibit a change in their health per their owner’s report. However, the appetite and hydration and cognitive functioning domains demonstrated somewhat poor reproducibility, thus highlighting these domains as potential candidates for removal from the measure. Test–retest reliability for the HCBQ was moderate overall, with ICCs between 0.70 and 0.79 (and 95% confidence interval lower bounds <0.70) demonstrated for the total score and the communication and quality-time domains. The HCBQ trust domain showed poor reproducibility; however, this domain was ultimately retained in the measure given that the remaining analyses indicated generally strong psychometric properties. Despite moderate support for test–retest reliability of both measures, the change in the scores was overall low between the two time points, potentially supporting natural variations in canine health and the HCB over a 2-week period.

Strong support was also found for the known-groups validity of the Canine HRQoL-Q and the HCBQ. The Canine HRQoL-Q was able to differentiate between groups as defined by canine health/disease states, with higher scores observed among healthy dogs, as expected. However, no significant differences were observed for the appetite and hydration, social functioning, and cognitive functioning domains, potentially suggesting that these domains may not be affected by the disease states included in our study. Nonetheless, significant differences were observed for all Canine HRQoL-Q domains by groups defined by the OGIH and canine general HRQoL/health, providing support for the known-groups validity overall.

Convergent validity, a measure of the degree to which concepts are related, was supported by moderate-to-strong and significant correlations between concepts, indicating that these concepts measure the same construct: the health-related quality of life of the dog. In addition, the Canine HRQoL-Q global score was found to be moderately, significantly, and positively correlated with the HCBQ total score (Pearson correlation coefficient = 0.44, *p* < 0.0001), demonstrating that there is convergent validity between the concepts of canine HRQoL and the concepts in the HCBQ. This finding indicates that the concepts of HRQoL and HCB are related, and a moderate correlation is to be expected if each measure also contributes uniquely to the concepts of interest. This relationship also suggests that the two measures together may measure a broader concept, e.g., the overall health and well-being of the dog. Further, this suggests that the availability of both measures in veterinary clinics can promote a broader, more comprehensive discussion about factors that influence a dog’s overall health, quality of life, and welfare than either measure alone.

Psychometric testing of the Canine HRQoL-Q and HCBQ indicated that some items might not reflect concepts that are prevalent enough among healthy dogs to warrant inclusion in a measure aimed at assessing generic HRQoL or the HCB. These concepts may have been identified during concept elicitation because only certain dogs specifically experienced them or were present in the relationships among only select dogs and their caretakers. Findings from psychometric analyses suggested that the deletion of some items/domains which are not adding informational content to the scales may improve the psychometric properties of each of the tools, leading to a final set of shorter questionnaires for ease of use in veterinarian clinics. As such, various items, as well as the underperforming appetite and hydration/social functioning domains, were removed from the measures. The final instrument questionnaires were retested and showed improved or similar psychometrics, providing support for the scoring algorithm. 

There are several strengths in the creation of the Canine HRQoL-Q and HCBQ. A major strength of this study was the expertise and experience of the research team in developing clinical outcomes assessments for use in human clinical trials or in human clinical practice. A rigorous development process using the principles and good research practices for development of outcomes measures outlined in FDA guidelines for the development of new drugs for humans was employed in developing the Canine HRQoL-Q and the HCBQ. Using this approach, the qualitative research conducted with caretakers of dogs provided an in-depth understanding of dog-specific HRQoL concepts from caretakers who are best positioned to assess the HRQoL of their dogs and who will complete the questionnaires in the veterinary office. Combining this data from caretakers with a literature review and data from interviews with veterinarians provided a rich foundation for the generation of items and the development of draft questionnaires. Piloting the draft questionnaires with caretakers to revise the questionnaire before quantitative testing was also a strength of the development process. Including the caretakers’ perspectives and language early on in the development process helped to ensure the relevancy, comprehensiveness, and ease of use of these questionnaires in veterinary practice.

While there is bias inherent in qualitative research, the research team designed the interview and focus group discussion guides to contain open-ended and non-leading questions to minimize bias in the data collection process. To minimize bias associated with the order of completion of the questionnaires in the cognitive interviews, the order was counterbalanced (i.e., the participant completed the HRQoL-Q first, followed by the HCBQ, or vice versa). To ensure the accuracy of and minimize bias associated with the summary notes for each part of qualitative data collection, the study members debriefed after the focus group or interview, and shared notes amongst the team to align on findings. Additionally, ensuring that content experts (i.e., veterinarians) were involved in the development process ensured that the concepts assessed in the measures were relevant to both caretakers and veterinarians. Furthermore, involving caretakers of dogs of various sociodemographic backgrounds and dogs of varying sizes, breeds, ages, and health statuses strengthened the content validation process.

The strengths of the psychometric validation study include the relatively large sample size (*N* = 327) recruited, which enabled the robust assessment of all the planned psychometric properties. The use of diversity quotas also ensured that the convenience sample is representative of the US population and the measurement properties of both tools are expected to be generalizable. An additional strength of the study relates to the development of the HCBQ to help understand the HRQoL of dogs. The HCBQ provides unique and important insights into concepts that potentially impact a dog’s HRQoL and have not been measured in the past for this purpose. To our knowledge, the HCBQ is the first questionnaire shown to be valid and reliable for assessing the bond between caretakers and their dogs.

Some limitations to this study should also be acknowledged. The participants in the qualitative research conducted to develop the draft questionnaires were a convenience sample recruited from a single, albeit large, clinical research company, which could potentially impact the generalizability and replicability of the analysis and study findings. Although efforts were made to recruit caretakers with a diverse sample of dogs, the sample may not represent the entire population of dogs (in terms of breed, size, age, and health status, etc.) or the caretakers of dogs. The quantitative testing was based on a non-interventional survey study design; thus, tests of sensitivity to change over time and the evaluation of clinically meaningful change thresholds were not possible. Further research is needed to determine whether the measures are responsive to change in a longitudinal study and what change in score reflects a change that is meaningful to veterinarians and canine caretakers. 

## 6. Conclusions

In conclusion, this study provides evidence in support of the reliability and the content and construct validity for the Canine HRQoL-Q and the HCBQ. As generic measures, the Canine HRQoL-Q and HCBQ can be used to reliably monitor the overall health and well-being of dogs in a veterinary practice and promote discussions and decision making between a veterinarian and caretaker over time. Used together, the two questionnaires provide a comprehensive understanding of factors contributing to the overall health and well-being of dogs. This comprehensiveness, coupled with the use of the principals outlined in FDA guidelines for the development of valid, reliable observer-reported outcome measures and the experience and expertise of the developers in the development of the Canine HRQoL-Q and HCBQ, provide an excellent option for use in veterinary practice. 

## Figures and Tables

**Table 1 animals-13-03255-t001:** Internal consistency reliability: Cronbach’s alpha and Cronbach’s alpha with each item deleted for the Canine HRQoL Questionnaire at visit 1.

Canine HRQOL Item	Total Score	Mobility	Energy and Vitality	Physical Health	Appetite and Hydration	Emotional Functioning	Cognitive Functioning	Social Functioning
My dog has difficulty moving around	0.95/0.96	0.85/0.85						
My dog has difficulty getting up	0.95/0.96	0.86/0.86						
My dog has difficulty walking	0.95/0.96	0.85/0.85						
My dog is able to jump	0.95/0.96	0.94/0.94						
My dog’s energy level is the same as usual	0.95/0.96		0.84/0.84					
My dog is lethargic	0.95/0.96		0.84/0.84					
My dog is fatigued	0.95/0.96		0.81/0.81					
My dog tires quickly	0.95/0.96		0.82/0.82					
My dog’s activity level is the same as usual	0.95/0.96		0.83/0.83					
My dog has skin irritation	0.96/0.96			0.82/0.82				
My dog vomits	0.96/0.96			0.83/0.83				
My dog has started having accidents	0.95/0.96			0.82/0.82				
My dog sleeps well	0.95/0.96			0.84/0.85				
My dog rests more than he/she normally does	0.95/0.96			0.81/0.81				
My dog vision seems to be getting worse	0.95/0.96			0.8/0.81				
My dog’s hearing seems to be getting worse	0.95/0.96			0.8/0.81				
My dog licks/chews/scratches himself excessively	0.95/0.96			0.81/0.82				
My dog is having difficulty chewing or swallowing food	0.95/0.96			0.82/0.81				
My dog has been hungrier than usual	0.95/0.96				0.74/0.77			
My dog has been less hungry than usual	0.95/0.96				0.78/0.78			
My dog has been drinking more water than usual	0.95/0.96				0.79/0.8			
My dog has been drinking less water than usual	0.95/0.96				0.75/0.75			
My dog does not enjoy his/her usual activity	0.95/0.96					0.75/0.77		
My dog seems happy	0.95/0.96					0.76/0.77		
My dog is playful	0.95/0.96					0.79/0.8		
My dog seems fearful	0.95/0.96					0.82/0.83		
My dog seems depressed	0.95/0.96					0.73/0.75		
My dog seems to be forgetting how to do things	0.95/0.96						0.88/0.89	
My dog sometimes seems confused	0.95/0.96						0.9/0.9	
My dog seems to be forgetting people and places	0.95/0.96						0.89/0.89	
My dog seems to get stuck in corners or behind objects	0.95/0.96						0.91/0.92	
My dog greets me when I have been away	0.95/0.96							0.45/0.46
My dog likes to play with other people/dog	0.95/0.96							0.57/0.58
My dog seems to want to be alone more than usual	0.95/0.96							0.51/0.54
Cronbach’s alpha (no item deleted)	0.95/0.96	0.91/0.91	0.86/0.86	0.83/0.84	0.81/0.82	0.81/0.82	0.92/0.92	0.6/0.62

**Table 2 animals-13-03255-t002:** Internal consistency reliability: Cronbach’s alpha and Cronbach’s alpha with each item deleted for HCBQ at visit 1.

HCBQ Item	Total Score	Trust Security	Communication	Quality Time
My dog checks on me routinely	0.92/0.93	0.83/0.85		
I check on my dog routinely	0.93/0.93	0.82/0.85		
My dog is comforted by my presence	0.92/0.93	0.81/0.82		
My dog loves to be with me	0.92/0.93	0.81/0.83		
My dog trusts me	0.92/0.93	0.82/0.83		
My dog usually seeks me out in new situations	0.93/0.94	0.87/0.88		
My dog likes positive reinforcement	0.93/0.93		0.73/0.76	
My dog makes eye contact with me	0.92/0.93		0.72/0.74	
My dog obeys my commands	0.93/0.94		0.81/0.81	
My dog and I communicate well with each other	0.92/0.93		0.69/0.72	
My dog wants to sit with me	0.93/0.93			0.85/0.85
My dog always wants me to pet him/her	0.93/0.94			0.85/0.86
My dog seeks my approval	0.93/0.93			0.84/0.85
My dog is included in my or my family’s activities	0.93/0.94			0.85/0.86
I make it a priority to spend quality time with my dog	0.92/0.93			0.84/0.84
My dog and I enjoy physical activities together	0.93/0.93			0.84/0.85
I consider my dog a member of my family	0.93/0.93			0.85/0.86
My dog is my companion	0.93/0.93			0.85/0.85
My dog seeks direction from me	0.93/0.93			0.84/0.85
Cronbach’s alpha (no item deleted)	0.93/0.94	0.85/0.87	0.79/0.81	0.86/0.87

**Table 3 animals-13-03255-t003:** Internal consistency reliability: item-total correlations for the Canine HRQoL Questionnaire at visit 1.

Canine HRQOL Item	Total Score	Mobility	Energy and Vitality	Physical Health	Appetite and Hydration	Emotional Functioning	Cognitive Functioning	Social Functioning
My dog has difficulty moving around	0.784 (<0.0001)	0.925 (<0.0001)						
My dog has difficulty getting up	0.746 (<0.0001)	0.911 (<0.0001)						
My dog has difficulty walking	0.784 (<0.0001)	0.93 (<0.0001)						
My dog is able to jump	0.556 (<0.0001)	0.773 (<0.0001)						
My dog’s energy level is the same as usual	0.592 (<0.0001)		0.77 (<0.0001)					
My dog is lethargic	0.714 (<0.0001)		0.769 (<0.0001)					
My dog is fatigued	0.769 (<0.0001)		0.833 (<0.0001)					
My dog tires quickly	0.713 (<0.0001)		0.832 (<0.0001)					
My dog’s activity level is the same as usual	0.611 (<0.0001)		0.786 (<0.0001)					
My dog has skin irritation	0.525 (<0.0001)			0.678 (<0.0001)				
My dog vomits	0.44 (<0.0001)			0.549 (<0.0001)				
My dog has started having accidents	0.587 (<0.0001)			0.644 (<0.0001)				
My dog sleeps well	0.46 (<0.0001)			0.421 (<0.0001)				
My dog rests more than he/she normally does	0.74 (<0.0001)			0.72 (<0.0001)				
My dog vision seems to be getting worse	0.75 (<0.0001)			0.761 (<0.0001)				
My dog’s hearing seems to be getting worse	0.72 (<0.0001)			0.744 (<0.0001)				
My dog licks/chews/scratches himself excessively	0.521 (<0.0001)			0.723 (<0.0001)				
My dog is having difficulty chewing or swallowing food	0.667 (<0.0001)			0.67 (<0.0001)				
My dog has been hungrier than usual	0.616 (<0.0001)				0.834 (<0.0001)			
My dog has been less hungry than usual	0.647 (<0.0001)				0.779 (<0.0001)			
My dog has been drinking more water than usual	0.643 (<0.0001)				0.81 (<0.0001)			
My dog has been drinking less water than usual	0.659 (<0.0001)				0.806 (<0.0001)			
My dog does not enjoy his/her usual activity	0.649 (<0.0001)					0.806 (<0.0001)		
My dog seems happy	0.626 (<0.0001)					0.771 (<0.0001)		
My dog is playful	0.677 (<0.0001)					0.712 (<0.0001)		
My dog seems fearful	0.486 (<0.0001)					0.679 (<0.0001)		
My dog seems depressed	0.687 (<0.0001)					0.836 (<0.0001)		
My dog seems to be forgetting how to do things	0.751 (<0.0001)						0.922 (<0.0001)	
My dog sometimes seems confused	0.77 (<0.0001)						0.908 (<0.0001)	
My dog seems to be forgetting people and places	0.728 (<0.0001)						0.911 (<0.0001)	
My dog seems to get stuck in corners or behind objects	0.711 (<0.0001)						0.868 (<0.0001)	
My dog greets me when I have been away	0.419 (<0.0001)							0.731 (<0.0001)
My dog likes to play with other people/dog	0.539 (<0.0001)							0.8 (<0.0001)
My dog seems to want to be alone more than usual	0.614 (<0.0001)							0.722 (<0.0001)

**Table 4 animals-13-03255-t004:** Internal consistency reliability: item-total correlations for the HCBQ at visit 1.

HCBQ Item	Total Score	Trust Security	Communication	Quality Time
My dog checks on me routinely	0.698 (<0.0001)	0.763 (<0.0001)		
I check on my dog routinely	0.691 (<0.0001)	0.761 (<0.0001)		
My dog is comforted by my presence	0.761 (<0.0001)	0.839 (<0.0001)		
My dog loves to be with me	0.761 (<0.0001)	0.801 (<0.0001)		
My dog trusts me	0.746 (<0.0001)	0.791 (<0.0001)		
My dog usually seeks me out in new situations	0.539 (<0.0001)	0.687 (<0.0001)		
My dog likes positive reinforcement	0.675 (<0.0001)		0.772 (<0.0001)	
My dog makes eye contact with me	0.758 (<0.0001)		0.786 (<0.0001)	
My dog obeys my commands	0.601 (<0.0001)		0.781 (<0.0001)	
My dog and I communicate well with each other	0.76 (<0.0001)		0.832 (<0.0001)	
My dog wants to sit with me	0.672 (<0.0001)			0.678 (<0.0001)
My dog always wants me to pet him/her	0.603 (<0.0001)			0.645 (<0.0001)
My dog seeks my approval	0.7 (<0.0001)			0.755 (<0.0001)
My dog is included in my or my family’s activities	0.608 (<0.0001)			0.678 (<0.0001)
I make it a priority to spend quality time with my dog	0.712 (<0.0001)			0.73 (<0.0001)
My dog and I enjoy physical activities together	0.684 (<0.0001)			0.733 (<0.0001)
I consider my dog a member of my family	0.603 (<0.0001)			0.594 (<0.0001)
My dog is my companion	0.686 (<0.0001)			0.682 (<0.0001)
My dog seeks direction from me	0.685 (<0.0001)			0.736 (<0.0001)

**Table 5 animals-13-03255-t005:** Test–retest reliability: mean scores and mean changes in the Canine HRQoL Questionnaire scores from visit 1 to visit 2 with ICCs for dogs defined as stable based on the OGIH.

Canine HRQoL Questionnaire Score	N	Mean (SD) at Visit 1	Mean (SD) at Visit 2	Mean Change (SD)	ICC (95% CI)
Canine HRQoL—Mobility subscore	54	8.65 (2.02)	8.52 (1.94)	−0.13 (1.09)	0.85 (0.75–0.91)
Canine HRQoL—Energy/vitality subscore	54	7.70 (2.13)	7.45 (1.99)	−0.25 (1.3)	0.80 (0.68–0.88)
Canine HRQoL—Physical health subscore	54	7.99 (1.60)	7.55 (1.71)	−0.44 (0.9)	0.85 (0.76–0.91)
Canine HRQoL—Appetite subscore	54	7.99 (1.79)	7.66 (1.88)	−0.32 (1.52)	0.66 (0.47–0.78)
Canine HRQoL–Emotional functioning subscore	54	8.34 (1.71)	7.99 (1.68)	−0.35 (1.11)	0.79 (0.66–0.87)
Canine HRQoL—Cognitive functioning subscore	54	8.91 (1.46)	8.65 (1.75)	−0.27 (1.35)	0.65 (0.46–0.78)
Canine HRQoL—Social functioning subscore	54	8.15 (1.72)	7.95 (1.80)	−0.20 (1.11)	0.80 (0.68–0.88)
Canine HRQoL—General health subscore	54	8.76 (1.36)	8.55 (1.32)	−0.21 (0.65)	0.88 (0.81–0.93)
Canine HRQoL—Total score	54	8.20 (1.47)	7.89 (1.54)	−0.31 (0.63)	0.91 (0.86–0.95)

**Table 6 animals-13-03255-t006:** Test–retest reliability: mean scores and mean changes in the HCBQ scores from visit 1 to visit 2 with ICCs for dogs defined as stable based on the OGIH.

HCBQ Score	N	Mean (SD) at Visit 1	Mean (SD) at Visit 2	Mean Change (SD)	ICC (95% CI)
HCBQ—Trust/security subscore	54	8.99 (1.11)	8.89 (1.32)	−0.10 (1.12)	0.58 (0.37–0.73)
HCBQ—Communication subscore	54	8.77 (1.21)	8.74 (1.19)	−0.03 (0.82)	0.77 (0.63–0.86)
HCBQ—Quality time subscore	54	8.77 (1.19)	8.61 (1.17)	−0.16 (0.83)	0.75 (0.61–0.85)
HCBQ—General bonding subscore	54	9.64 (0.66)	9.48 (0.80)	−0.16 (0.56)	0.70 (0.54–0.82)
HCBQ—Total score	54	8.84 (1.06)	8.72 (1.10)	−0.12 (0.75)	0.76 (0.62–0.85)

**Table 7 animals-13-03255-t007:** Known-groups validity: comparison of the Canine HRQoL Questionnaire scores between severity groups defined by the dog’s health/disease state, age, OGIH and general health items at visit 1.

Canine HRQoL Questionnaire Score	Group Definition	*n*	Mean (SD)	*p*-Value	Effect Size, ɳ^2^
Mobility	Canine health/disease states				
	Healthy	204	8.76 (1.79)		
	Non-food allergy/skin problems	62	7.88 (2.64)	<0.0001	0.07
	Ear infection	16	6.76 (2.85)		
	Age				
	Tertile 1 (≤4.5 years)	109	8.76 (1.99)		
	Tertile 2 (4.5–8.5 years)	110	8.91 (1.61)	<0.0001	0.10
	Tertile 3 (≥8.5 years)	108	7.31 (2.72)		
	OGIH				
	Excellent	96	9.27 (1.65)		
	Very good	147	8.79 (1.73)	<0.0001	0.31
	Good	64	7.13 (2.17)		
	Fair/poor	20	4.34 (2.86)		
	Canine HRQoL questionnaire—item 8a				
	Tertile 1 (<8)	76	6.23 (2.73)		
	Tertile 2 (8–9)	173	8.72 (1.75)	<0.0001	0.28
	Tertile 3 (10)	78	9.51 (1.25)		
	Canine HRQoL questionnaire—item 8b				
	Tertile 1 (<9)	95	6.88 (2.71)		
	Tertile 2 (9)	86	8.31 (1.97)	<0.0001	0.20
	Tertile 3 (10)	146	9.29 (1.48)		
Energy and vitality	Canine health/disease states				
	Healthy	204	8.10 (1.73)		
	Non-food allergy/skin problems	62	7.15 (2.28)	<0.0001	0.08
	Ear infection	16	6.13 (2.76)		
	Age				
	Tertile 1 (≤4.5 years)	109	8.22 (2.00)		
	Tertile 2 (4.5–8.5 years)	110	8.00 (1.74)	<0.0001	0.10
	Tertile 3 (≥8.5 years)	108	6.70 (2.18)		
	OGIH				
	Excellent	96	8.82 (1.80)		
	Very good	147	7.82 (1.64)	<0.0001	0.33
	Good	64	6.60 (1.79)		
	Fair/poor	20	4.08 (1.62)		
	Canine HRQoL questionnaire—item 8a				
	Tertile 1 (<8)	76	5.68 (2.01)		
	Tertile 2 (8–9)	173	7.93 (1.67)	<0.0001	0.31
	Tertile 3 (10)	78	8.94 (1.61)		
	Canine HRQoL questionnaire—item 8b				
	Tertile 1 (<9)	95	6.25 (2.08)		
	Tertile 2 (9)	86	7.55 (1.74)	<0.0001	0.23
	Tertile 3 (10)	146	8.62 (1.72)		
Physical health	Canine health/disease states				
	Healthy	204	8.16 (1.67)		
	Non-food allergy/skin problems	62	6.55 (1.52)	<0.0001	0.14
	Ear infection	16	7.36 (1.80)		
	Age				
	Tertile 1 (≤4.5 years)	109	7.97 (2.00)		
	Tertile 2 (4.5–8.5 years)	110	7.87 (1.53)	<0.0001	0.08
	Tertile 3 (≥8.5 years)	108	6.83 (1.84)		
	OGIH				
	Excellent	96	8.61 (1.78)		
	Very good	147	7.74 (1.46)	<0.0001	0.30
	Good	64	6.41 (1.48)		
	Fair/poor	20	4.89 (1.52)		
	Canine HRQoL questionnaire—item 8a				
	Tertile 1 (<8)	76	6.02 (1.63)		
	Tertile 2 (8–9)	173	7.71 (1.57)	<0.0001	0.26
	Tertile 3 (10)	78	8.74 (1.69)		
	Canine HRQoL questionnaire—item 8b				
	Tertile 1 (<9)	95	6.36 (1.82)		
	Tertile 2 (9)	86	7.52 (1.58)	<0.0001	0.21
	Tertile 3 (10)	146	8.37 (1.62)		
Appetite and hydration	Canine health/disease states				
	Healthy	204	8.09 (1.93)		
	Non-food allergy/skin problems	62	7.71 (1.75)	0.35	0.01
	Ear infection	16	7.81 (1.44)		
	Age				
	Tertile 1 (≤4.5 years)	109	7.99 (2.18)		
	Tertile 2 (4.5–8.5 years)	110	8.10 (1.69)	0.06	0.02
	Tertile 3 (≥8.5 years)	108	7.53 (1.73)		
	OGIH				
	Excellent	96	8.72 (1.88)		
	Very good	147	8.04 (1.61)	<0.0001	0.19
	Good	64	6.82 (1.81)		
	Fair/poor	20	5.97 (1.22)		
	Canine HRQoL questionnaire—item 8a				
	Tertile 1 (<8)	76	6.61 (1.68)		
	Tertile 2 (8–9)	173	7.95 (1.75)	<0.0001	0.18
	Tertile 3 (10)	78	8.93 (1.69)		
	Canine HRQoL questionnaire—item 8b				
	Tertile 1 (<9)	95	6.60 (1.68)		
	Tertile 2 (9)	86	7.89 (1.79)	<0.0001	0.22
	Tertile 3 (10)	146	8.69 (1.61)		
Emotional functioning	Canine health/disease states				
	Healthy	204	8.52 (1.59)		
	Non-food allergy/skin problems	62	7.85 (1.59)	0.01	0.03
	Ear infection	16	8.16 (1.48)		
	Age				
	Tertile 1 (≤4.5 years)	109	8.34 (1.96)		
	Tertile 2 (4.5–8.5 years)	110	8.61 (1.30)	0.002	0.04
	Tertile 3 (≥8.5 years)	108	7.83 (1.56)		
	OGIH				
	Excellent	96	9.16 (1.47)		
	Very good	147	8.43 (1.37)	<0.0001	0.30
	Good	64	7.30 (1.35)		
	Fair/poor	20	5.78 (1.39)		
	Canine HRQoL questionnaire—item 8a				
	Tertile 1 (<8)	76	6.87 (1.56)		
	Tertile 2 (8–9)	173	8.45 (1.39)	<0.0001	0.25
	Tertile 3 (10)	78	9.22 (1.42)		
	Canine HRQoL questionnaire—item 8b				
	Tertile 1 (<9)	95	7.04 (1.55)		
	Tertile 2 (9)	86	8.15 (1.46)	<0.0001	0.28
	Tertile 3 (10)	146	9.13 (1.28)		
Cognitive functioning	Canine health/disease states				
	Healthy	204	8.92 (1.76)		
	Non-food allergy/skin problems	62	8.51 (2.16)	0.26	0.01
	Ear infection	16	8.59 (1.34)		
	Age				
	Tertile 1 (≤4.5 years)	109	8.70 (2.19)		
	Tertile 2 (4.5–8.5 years)	110	9.13 (1.23)	0.004	0.03
	Tertile 3 (≥8.5 years)	108	8.25 (2.20)		
	OGIH				
	Excellent	96	9.28 (1.92)		
	Very good	147	9.06 (1.35)	<0.0001	0.17
	Good	64	7.64 (2.13)		
	Fair/poor	20	6.63 (2.58)		
	Canine HRQoL questionnaire—item 8a				
	Tertile 1 (<8)	76	7.40 (2.29)		
	Tertile 2 (8–9)	173	9.02 (1.50)	<0.0001	0.14
	Tertile 3 (10)	78	9.23 (1.94)		
	Canine HRQoL questionnaire—item 8b				
	Tertile 1 (<9)	95	7.41 (2.23)		
	Tertile 2 (9)	86	8.88 (1.59)	<0.0001	0.19
	Tertile 3 (10)	146	9.42 (1.49)		
Social functioning	Canine health/disease states				
	Healthy	204	8.40 (1.64)		
	Non-food allergy/skin problems	62	7.97 (2.26)	0.21	0.01
	Ear infection	16	8.00 (1.82)		
	Age				
	Tertile 1 (≤4.5 years)	109	8.51 (1.70)		
	Tertile 2 (4.5–8.5 years)	110	8.61 (1.49)	<0.001	0.06
	Tertile 3 (≥8.5 years)	108	7.69 (1.79)		
	OGIH				
	Excellent	96	9.14 (1.41)		
	Very good	147	8.35 (1.51)	<0.0001	0.23
	Good	64	7.50 (1.53)		
	Fair/poor	20	5.96 (1.84)		
	Canine HRQoL questionnaire—item 8a				
	Tertile 1 (<8)	76	7.06 (1.76)		
	Tertile 2 (8–9)	173	8.40 (1.54)	<0.0001	0.18
	Tertile 3 (10)	78	9.16 (1.35)		
	Canine HRQoL questionnaire—item 8b				
	Tertile 1 (<9)	95	7.20 (1.79)		
	Tertile 2 (9)	86	8.28 (1.49)	<0.0001	0.19
	Tertile 3 (10)	146	8.96 (1.40)		
Total score	Canine health/disease states				
	Healthy	204	8.38 (1.41)		
	Non-food allergy/skin problems	62	7.48 (1.59)	<0.0001	0.07
	Ear infection	16	7.48 (1.61)		
	Age				
	Tertile 1 (≤4.5 years)	109	8.29 (1.75)		
	Tertile 2 (4.5–8.5 years)	110	8.36 (1.21)	<0.0001	0.09
	Tertile 3 (≥8.5 years)	108	7.34 (1.57)		
	OGIH				
	Excellent	96	8.94 (1.42)		
	Very good	147	8.22 (1.14)	<0.0001	0.38
	Good	64	6.94 (1.27)		
	Fair/poor	20	5.26 (1.23)		
	Canine HRQoL questionnaire—item 8a				
	Tertile 1 (<8)	76	6.44 (1.47)		
	Tertile 2 (8–9)	173	8.21 (1.24)	<0.0001	0.34
	Tertile 3 (10)	78	9.05 (1.26)		
	Canine HRQoL questionnaire—item 8b				
	Tertile 1 (<9)	95	6.73 (1.57)		
	Tertile 2 (9)	86	7.98 (1.26)	<0.0001	0.31
	Tertile 3 (10)	146	8.84 (1.18)		

**Table 8 animals-13-03255-t008:** Known-groups validity: comparison of the Canine HRQoL Questionnaire scores between severity groups defined by the dog’s health/disease state, age, OGIH and general bonding item at visit 1.

HCBQ Score	Group Definition	*n*	Mean (SD)	*p*-Value	Effect Size, ɳ^2^
Trust and security	Canine health/disease states				
	Healthy	204	8.87 (1.24)		
	Non-food allergy/skin problems	62	9.13 (1.04)	0.19	0.01
	Ear infection	16	9.24 (0.95)		
	Age				
	Tertile 1 (≤4.5 years)	109	8.96 (1.33)		
	Tertile 2 (4.5–8.5 years)	110	9.07 (1.01)	0.46	0.00
	Tertile 3 (≥8.5 years)	108	8.87 (1.18)		
	OGIH				
	Excellent	96	9.47 (0.90)		
	Very good	147	9.01 (1.10)	<0.0001	0.13
	Good	64	8.31 (1.34)		
	Fair/poor	20	8.40 (1.24)		
	HCBQ—item 4a				
	Below median (≤9)	115	8.12 (1.28)	<0.0001	0.28
	Above median (10)	212	9.43 (0.81)		
	HCBQ—item 4b				
	Below median (≤9)	103	8.10 (1.29)	<0.0001	0.25
	Above median (10)	224	9.37 (0.87)		
Communication	Canine health/disease states				
	Healthy	204	8.73 (1.31)		
	Non-food allergy/skin problems	62	8.48 (1.24)	0.31	0.01
	Ear infection	16	8.91 (0.81)		
	Age				
	Tertile 1 (≤4.5 years)	109	8.74 (1.39)		
	Tertile 2 (4.5–8.5 years)	110	8.80 (1.14)	0.23	0.01
	Tertile 3 (≥8.5 years)	108	8.51 (1.36)		
	OGIH				
	Excellent	96	9.45 (0.90)		
	Very good	147	8.69 (1.18)	<0.0001	0.22
	Good	64	7.78 (1.30)		
	Fair/poor	20	7.88 (1.58)		
	HCBQ—item 4a				
	Below median (≤9)	115	7.83 (1.35)	<0.0001	0.24
	Above median (10)	212	9.15 (1.01)		
	HCBQ—item 4b				
	Below median (≤9)	103	7.78 (1.34)	<0.0001	0.22
	Above median (10)	224	9.10 (1.05)		
Spending quality time/companionship	Canine health/disease states				
	Healthy	204	8.60 (1.32)		
	Non-food allergy/skin problems	62	8.52 (1.18)	0.87	0.00
	Ear infection	16	8.47 (1.22)		
	Age				
	Tertile 1 (≤4.5 years)	109	8.80 (1.29)		
	Tertile 2 (4.5–8.5 years)	110	8.73 (1.05)	0.002	0.04
	Tertile 3 (≥8.5 years)	108	8.23 (1.43)		
	OGIH				
	Excellent	96	9.28 (0.94)		
	Very good	147	8.64 (1.11)	<0.0001	0.21
	Good	64	8.64 (1.11)		
	Fair/poor	20	7.78 (1.55)		
	HCBQ—item 4a				
	Below median (≤9)	115	7.67 (1.28)	<0.0001	0.27
	Above median (10)	212	9.08 (0.99)		
	HCBQ—item 4b				
	Below median (≤9)	103	7.62 (1.23)	<0.0001	0.26
	Above median (10)	224	9.03 (1.05)		
Total score	Canine health/disease states				
	Healthy	204	8.71 (1.19)		
	Non-food allergy/skin problems	62	8.70 (1.02)	0.95	0.00
	Ear infection	16	8.81 (0.95)		
	Age				
	Tertile 1 (≤4.5 years)	109	8.84 (1.24)		
	Tertile 2 (4.5–8.5 years)	110	8.85 (0.94)	0.03	0.02
	Tertile 3 (≥8.5 years)	108	8.49 (1.22)		
	OGIH				
	Excellent	96	9.38 (0.82)		
	Very good	147	8.76 (1.00)	<0.0001	0.22
	Good	64	7.90 (1.23)		
	Fair/poor	20	7.99 (1.32)		
	HCBQ—item 4a				
	Below median (≤9)	115	7.85 (1.19)	<0.0001	0.32
	Above median (10)	212	9.20 (0.80)		
	HCBQ—item 4b				
	Below median (≤9)	103	7.81 (1.16)	<0.0001	0.29
	Above median (10)	224	9.15 (0.87)		

**Table 9 animals-13-03255-t009:** Convergent validity: correlations among the Canine HRQoL Questionnaire total and HRQoL domain scores at visit 1.

Domain	Mobility	Energy and Vitality	Physical Health	Appetite	Emotional Functioning	Cognitive Functioning	Social Functioning	General Health	Total Score
Mobility subscore	1.00 (<0.0001)	0.73 (<0.0001)	0.65 (<0.0001)	0.51 (<0.0001)	0.58 (<0.0001)	0.64 (<0.0001)	0.52 (<0.0001)	−0.62 (<0.0001)	0.81 (<0.0001)
Energy/vitality subscore	0.73 (<0.0001)	1.00 (<0.0001)	0.68 (<0.0001)	0.57 (<0.0001)	0.66 (<0.0001)	0.63 (<0.0001)	0.59 (<0.0001)	−0.62 (<0.0001)	0.85 (<0.0001)
Physical health subscore	0.65 (<0.0001)	0.68 (<0.0001)	1.00 (<0.0001)	0.70 (<0.0001)	0.66 (<0.0001)	0.72 (<0.0001)	0.52 (<0.0001)	−0.60 (<0.0001)	0.90 (<0.0001)
Appetite subscore	0.51 (<0.0001)	0.57 (<0.0001)	0.70 (<0.0001)	1.00 (<0.0001)	0.67 (<0.0001)	0.62 (<0.0001)	0.52 (<0.0001)	−0.50 (<0.0001)	0.79 (<0.0001)
Emotional functioning subscore	0.58 (<0.0001)	0.66 (<0.0001)	0.66 (<0.0001)	0.67 (<0.0001)	1.00 (<0.0001)	0.58 (<0.0001)	0.62 (<0.0001)	−0.59 (<0.0001)	0.82 (<0.0001)
Cognitive functioning subscore	0.64 (<0.0001)	0.63 (<0.0001)	0.72 (<0.0001)	0.62 (<0.0001)	0.58 (<0.0001)	1.00 (<0.0001)	0.53 (<0.0001)	−0.51 (<0.0001)	0.82 (<0.0001)
Social functioning subscore	0.52 (<0.0001)	0.59 (<0.0001)	0.52 (<0.0001)	0.52 (<0.0001)	0.62 (<0.0001)	0.53 (<0.0001)	1.00 (<0.0001)	−0.48 (<0.0001)	0.70 (<0.0001)
General health subscore	−0.62 (<0.0001)	−0.62 (<0.0001)	−0.60 (<0.0001)	−0.50 (<0.0001)	−0.59 (<0.0001)	−0.51 (<0.0001)	−0.48 (<0.0001)	1.00 (<0.0001)	−0.69 (<0.0001)
Total score	0.81 (<0.0001)	0.85 (<0.0001)	0.90 (<0.0001)	0.79 (<0.0001)	0.82 (<0.0001)	0.82 (<0.0001)	0.70 (<0.0001)	−0.69 (<0.0001)	1.00 (<0.0001)

**Table 10 animals-13-03255-t010:** Convergent validity: correlations among the HCBQ total and HCBQ domain scores at visit 1.

Domain	Trust Security	Communication	Quality Time	General Bonding	Total Score
Trust/security subscore	1.00 (<0.0001)	0.74 (<0.0001)	0.75 (<0.0001)	0.61 (<0.0001)	0.90 (<0.0001)
Communication subscore	0.74 (<0.0001)	1.00 (<0.0001)	0.75 (<0.0001)	0.55 (<0.0001)	0.87 (<0.0001)
Quality time subscore	0.75 (<0.0001)	0.75 (<0.0001)	1.00 (<0.0001)	0.60 (<0.0001)	0.95 (<0.0001)
General bonding subscore	0.61 (<0.0001)	0.55 (<0.0001)	0.60 (<0.0001)	1.00 (<0.0001)	0.64 (<0.0001)
Total score	0.90 (<0.0001)	0.87 (<0.0001)	0.95 (<0.0001)	0.64 (<0.0001)	1.00 (<0.0001)

## Data Availability

Relevant data generated during the tool development phase are included in this article. The scoring algorithm is not presented here to preserve intellectual property rights.

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
