# Peer review of "Development and Validation of a Canine Health-Related Quality of Life Questionnaire and a Human–Canine Bond Questionnaire for Use in Veterinary Practice"

_animals, 2023, doi:10.3390/ani13203255_

Round 1
Author Response
Response to Reviewer 1.
9/19/23
From Robert Lavan, Muna Tahir, Christina O’Donnell, Alex Bellinger, Elodie DeBock and Pat Koochoki
Summary:
The aim of this study was to develop and validate two questionnaires relating to 1) health- related canine quality of life and 2) the canine human bond. These questionnaires were developed via the use of collages, interviews and psychometric testing. It is hoped that these questionnaires may be of use to assess the overall health and well-being of dogs in a veterinary clinic setting.
Specific comments (please note, line numbers were not included on the version provided):
- 2.1.1 Paragraph Two - “Photo collages are a projective technique in which participants project their opinions and beliefs onto images that they select about a particular topic. This allows participants to articulate their thoughts and experiences at a subconscious level that is difficult to reach with direct questioning techniques frequently used in qualitative interviewing.” Can you please add a reference to support this technique.
Response from the Authors: Three references have been added to the manuscript.
- 2.1.1. - “The participants were also asked to complete the published 15-item canine HRQoL survey (CHQLS-15) and to provide perspective on the relevance and importance of the concepts addressed in the questionnaire.” I find this sentence hard to follow as my understanding was that the HRQoL scale was not yet published, and that during this stage it was not yet even finalized. Can you please add clarity to this sentence.
Response from the Authors: In 2013, Dr. Robert Lavan published the CHQLS-151 as his first attempt to assess the Quality of Life of dogs that owners thought were essentially healthy. When we decided to create a new QoL measure, we thought we could do better and wanted to start by seeing how the items and domains in this earlier measure stacked up against the items and domains we were assessing in the qualitative stage of the current tool.
1 Lavan RP. Development and validation of a survey for quality of life assessment by owners of healthy dogs. The Veterinary Journal. (2013) 197: 578-582. Reference added to the manuscript.
- 2.2.1 - Can you please add details on how these participants were selected? It says that “interested participants received an email…” but it is not clear how these people were first found or how they expressed their interest to you.
Response from the Authors: Participants for concept elicitation focus groups conducted to develop the initial draft of the instruments was a convenience sample of employee volunteers from ICON plc. Cognitive interviews to establish content validity and the quantitative psychometric testing were conducted with canine caretakers recruited by a market research agency.
- 2.3 Statistical Analysis - First sentence. Please add more details on the pre-specified statistical analysis plan. If this is what is being described in 2.3.1 etc., please direct the reader using a phrase such as “outlined below” so that it is clear that you will be describing the plan.
Response from the Authors: The sentence on pre-specified statistical analysis plan has been removed as it is not needed and seems to cause confusion. The reader will instead be directed to Sections 2.3.1-2.3.4
- 4.2.1 Results - I would like to see here how many participants started and how many participants completed the online survey, and how you dealt with surveys that were started but not completed.
Response from the Authors: The number of participants who started the survey but did not complete it, is unknown. A total of 327 participants completed visit 1 of the online survey, while 75 participants completed visit 1 and 2 online surveys (to determine test-retest reliability). As stated in the manuscript, “If a respondent did not complete the questionnaire, responses were not used.”
General comments:
- The paper describes a literature review as the first step of the process (both in the introduction and discussion) however in both 2.1 (materials and methods) and 4.1 (results) there is no mention of 1) how this review was conducted or 2) what was found. Please add more details on this as it is mentioned throughout.
Response from the Authors: Text detailing the literature review methods and results has been added to the manuscript.
- In general, the CHQLS-15 requires more explanation. When searching the term, it appears only twice and neither time does it have adequate explanation on what it is and how it fits in with the draft survey. This ties into my previous comment made about section 2.1.1.
Response from the Authors: We added text in the Material and Methods, 3rd paragraph under 2.1.1 to explain the use of this prior survey tool. A reference was also added.
- It is currently unclear whether OGIH something that exists/has been used in previous literature. Please add more detail when it is first mentioned (in section 1.2.2.1). If it is an existing metric, please cite appropriately. If novel, please add why you wanted to include it and how it is distinct from your other questions.
Response from the Authors: The OGIH (Owner Global Impression of Health) is a single-item measure that assesses a caregiver's impression of their dog's general health. Specifically, caregivers are asked to rate the health of their dog as follows: "Excellent," "Very good," "Good," "Fair," and "Poor." Higher scores denote the worse health of the dog. We developed this measure to use it as a CGI (Clinical Global Impression scale), which was published in 1976 by the National Institute of Mental Health (US). It consists of one item that mainly measures change in clinical status. Over the years, CGI and derived scales of CGI were used in a broad range of diseases and were modified for the purpose of clinical settings (item label, number of response options and response options). This is a standard practice to develop and use this kind of measure in order to assess a global impression of the concept of interest and then use this scale’s assessment as an anchor for psychometric validation of an instrument (as we did within this study). This helps psychometricians to define stable groups between two time points (for test-retest reliability assessment) and also to define groups at a specific time point (for known-groups validity assessment).
- More background information and rationale are needed when it is stated that this survey is intended for “healthy dogs” (e.g., in section 2.3.4). If the surveys’ intended for uses are a monitoring system across time, will it not eventually be being filled out for increasingly “unhealthy” dogs? This is important as it has led to some decisions, such as the removal of items like “underperforming appetite” and “hydration/ social functioning” domains. While these may not be relevant for initially healthy dogs, they might be very important to keep track of if using as described as a monitoring system of health across time?
Response from the Authors: The concept of healthy dogs in this study stems from the fact that primarily, we used dogs that owners would self-classify as healthy. The veterinary clinical definition of healthy might involve ensuring blood work with all values within normal ranges and/or normal appearing radiographs and/or have had an unremarkable physical exam (no lumps, bumps or abnormalities noted). The quality of life assessment does not take the veterinarian opinion into account. The QoL measure is an assessment of the owner’s opinion of how the dog is doing now. To that end, an owner might consider a diabetic dog healthy if they are well controlled by medication. We also agree that the degree of health of the dog will change over time, even if no new disease appears, just because of natural aging. This tool accepts that the health quality changes over time (both positively and negatively) and that it may become less sensitive as the animal ages. This is part of the training we will offer veterinarians and pet owners when this becomes publicly accessible. Over several year of examining possible concepts for inclusion, we have come to realize that some items don’t differentiate between sick and healthy even though it might initially seem like they should. Water and food consumption are good examples. All very ill dogs do not stop eating or drinking. Some do, some don’t. Some dogs, in the face of pain, continue to eat amazingly well. Dog owners who use food consumption as a measure of suffering will fail to see the need for intervention when the recommendation comes from the veterinarian. This is why some items don’t differentiate enough to be included.
- In the discussion, it would be interesting to add more on how these surveys fit in with existing surveys measuring canine quality of life and health related decline, for example the DISHAA survey for cognitive decline. More on 1) the intersection with pre-existing surveys 2) the relationship between physical and behavioral health components would be useful in the discussion.
Response from the Authors: Text describing and comparing existing questionnaires with the Canine HRQoL and HCBQ questionnaires, as well as the relationship between physical and behavioral health components have been added.
- Given the acknowledged limitation that you cannot currently assess sensitivity to change over time, I would perhaps re-phrase the first sentence of the conclusion where it states that the surveys can be used to “reliably monitor overall health and well-being of dogs in veterinary practice over time”. For this assertion to be made we would first need to test this formally with a longitudinal study. Perhaps this could be suggested as a future direction prior to the conclusion paragraph.
Response from the Authors: We agree and have modified the conclusion section to align with the solution proposed by the reviewer.
- Overall, I think that this is an interesting and largely well-written paper. However, there appears to be some key information missing or unclear, and the discussion lacks breadth. Once these issues have been resolved I believe that this paper will be suitable for publication.
Reviewer 2 Report
Dear Authors,
Thank you for a very well written manuscript. It highlights the need to have a validated method to assess the health related quality of life and how it affects human animal bond. These parameters are not easy to measure, and the reliability of owner assessment may also be questioned by the regulatory authorities. It is thus a very good starting point to use a guideline that already exists in the human side to create something relevant to animal health as well. However, developing something to the standards required in human medicine may not always be cost effective in the animal health side but this comment may not be so relevant in this case.
As the study was conducted in the US, there may be some cultural limitations regarding generalisation over the dog owner population in Europe or other parts of the world.
“While some generic measures of canine HRQoL do exist [12–14], they do not appear to follow all of the principles or best research practices in the regulatory guidance documents for the development of valid and reliable HRQoL measures.” This sentence may need some further clarification as there is a commercially available and validated HRQoL instrument: NewMetrica
Specific comments:
2.1.1. Concept Elicitation, Item Generation, Draft Questionnaire Development “Four virtual focus groups with approximately five participants each (20 total)” – were these the same dog owners you mention in the beginning of the chapter?
2.2.1 Study population: “Participants included caregivers of dogs for at least six months who were at least 18 years of age, English-speaking, located in the U.S and able to provide consent to participate in the study. Dogs were excluded from the study if they were fostered.” - The participants were people, please consider rewording this underlined sentence.
“Participants who were followed-up were only required to complete the questionnaires at the second time point/visit” – is it explained somewhere which participants were followed-up and why?
How many dogs did the participants own- only one or were they allowed to own several dogs?
2.3.1. Item-Level Analyses
“Item-level analyses were evaluated using visit 1 data” - please explain somewhere what is visit 1 and how many visits did the study participants have?
2.3.3. Reliability
“Test-retest reliability was assessed between visits 1 and 2 among dogs whose owners reported no change on the OGIH (i.e., stable dogs)” – what was the interval between visits 1 and 2?
Table 3
“My dog has started having accidents” – this is not quite clear to me. Does it mean that the dog has urinary incontinence or poops indoors, or something else?
Table 7:
It would be helpful if HRQoL items 8 a&b and 4 a&b were explained in the title or somewhere as now the table is not self explanatory, nor could I find explanations to these items in the text.
Health status of the dogs: Please justify why dogs were classified in these two groups: Non-food allergy/skin problems and ear infection as these conditions quite frequently co-occur? On the other hand, 20 dogs were assessed as being fair/poor in their general health status – what other diseases did they have?
What was the correlation between OGIH and canine general HRQoL/health?
Discussion:
“Similarly, HCBQ scores were not significantly different by canine health/disease state, as the HCB may not be impacted by condition, and indeed may even be stronger among sick dogs. However, total/domain scores differed significantly by the OGIH and general bonding items, with higher scores observed among dogs whose owners reported that their dog was in “excellent” health and whose owners reported a higher degree of bonding with their dog as anticipated.” These two underlined sentences are somewhat contradictory, if I understand this correctly.
Author Response
Dear Authors,
Thank you for a very well written manuscript. It highlights the need to have a validated method to assess the health related quality of life and how it affects human animal bond. These parameters are not easy to measure, and the reliability of owner assessment may also be questioned by the regulatory authorities. It is thus a very good starting point to use a guideline that already exists in the human side to create something relevant to animal health as well. However, developing something to the standards required in human medicine may not always be cost effective in the animal
As the study was conducted in the US, there may be some cultural limitations regarding generalisation over the dog owner population in Europe or other parts of the world.
“While some generic measures of canine HRQoL do exist [12–14], they do not appear to follow all of the principles or best research practices in the regulatory guidance documents for the development of valid and reliable HRQoL measures.” This sentence may need some further clarification as there is a commercially available and validated HRQoL instrument: NewMetrica
Response from the Authors: The sentence was modified per comment.
Specific comments:
2.1.1. Concept Elicitation, Item Generation, Draft Questionnaire Development “Four virtual focus groups with approximately five participants each (20 total)” – were these the same dog owners you mention in the beginning of the chapter?
Response from the Authors: Yes, this is correct.
2.2.1 Study population: “Participants included caregivers of dogs for at least six months who were at least 18 years of age, English- speaking, located in the U.S and able to provide consent to participate in the study. Dogs were excluded from the study if they were fostered.” - The participants were people, please consider rewording this underlined sentence.
Response from the Authors: The wording of this sentence has been revised.
“Participants who were followed-up were only required to complete the questionnaires at the second time point/visit” – is it explained somewhere which participants were followed-up and why?
Response from the authors: A subset of participants completed the questionnaire a second time to determine interrater reliability.
How many dogs did the participants own- only one or were they allowed to own several dogs?
Response from the Authors: Owners could have multiple dogs but were asked to focus on only one of their dogs for the purposes of the research.
2.3.1. Item-Level Analyses
“Item-level analyses were evaluated using visit 1 data” - please explain somewhere what is visit 1 and how many visits did the study participants have?
Response from the Authors: A subset of patients (n=70) completed the survey a second time (v2, two weeks after initially completing the survey at v1) to determine the reliability of the questionnaires over time. Responses at v2 were correlated with responses from v1 for those participants who reported no change in the health status of their dogs. If the health status of the dog did not change over the two weeks, little or no change is expected in other scores.
2.3.3. Reliability
“Test-retest reliability was assessed between visits 1 and 2 among
dogs whose owners reported no change on the OGIH (i.e., stable dogs)” – what
was the interval between visits 1 and 2?
Response from the authors: The interval between v1 and v2 was two weeks.
Table 3
“My dog has started having accidents” – this is not quite clear to me. Does it mean that the dog has urinary incontinence or poops indoors, or something else?
Response from the Authors: The item is intended to include with urinary incontinence or poops indoors. This item was cognitively debriefed during development of the questionnaire, none of the participants reported having difficulty understanding the meaning of the item.
Table 7:
It would be helpful if HRQoL items 8 a&b and 4 a&b were explained in the title or somewhere as now the table is not self explanatory, nor could I find explanations to these items in the text.
Response from the Authors: These are general items in the questionnaires. Additional wording is included in tables where these appear.
Health status of the dogs: Please justify why dogs were classified in these two groups: Non-food allergy/skin problems and ear infection as these conditions quite frequently co-occur? On the other hand, 20 dogs were assessed as being fair/poor in their general health status – what other diseases did they have?
Response from the Authors: The first part of the question involves section 2.3.4 Validity, where known-groups validity was examined at visit 1 to determine whether the canine HRQoL-Q and HCBQ can distinguish between groups with expected differences in scores. Pet owners were asked to assess their dog for health and were included if they stated that their dog was in good health. A small group of pet owners were also included whose dogs were reported to have atopy and/or ear infections. Although the questionnaires are intended to be used with healthy dogs, recruitment for this study included dogs with variable health statuses as reported by their caregivers, e.g., those that are considered healthy by their caregivers as well as dogs that are reported to be unwell (specifically those with atopy and ear infections) to assess this psychometric property. We did not ask about other diseases. We were more concerned about the owner’s overall assessment of dog health.
What was the correlation between OGIH and canine general
HRQoL/health?
Response from the Authors: We did not use the OGIH instrument to correlate directly with general HRQoL/health, we used it to define groups that make sense to assess the known-groups validity of the HRQoL questionnaire and also to perform the test-retest reliability on stable dogs (group of stable dogs assessed by OGIH between visits 1 and 2). A correlation between OGIH and canine general HRQoL/health would have been a strong correlation. Indeed, this is expected when we see the results obtained for known-groups validity. When canine HRQoL questionnaire scores were compared across OGIH group categories and by the general health items in the questionnaire, it was evident that dogs whose owners reported them being in “excellent” health and those with better health and a higher quality of life had significantly lower (better) scores, compared to those in “poor/fair” health and those with poor health and a poor quality of life.
Discussion:
“Similarly, HCBQ scores were not significantly different by canine health/disease state, as the HCB may not be impacted by condition, and indeed may even be stronger among sick dogs. However, total/domain scores differed significantly by the OGIH and general bonding items, with higher scores observed among dogs whose owners reported that their dog was in “excellent”
health and whose owners reported a higher degree of bonding with their dog as anticipated.” These two underlined sentences are
somewhat contradictory, if I understand this correctly.
Response from the Authors: The Authors agree that this paragraph is somewhat confusing, and that the information does not add very much to the discussion of the questionnaires. Therefore, this paragraph has been deleted from the manuscript.